# CRNet: Image Super-Resolution Using A Convolutional Sparse Coding Inspired Network

## Abstract

Convolutional Sparse Coding (CSC) has been attracting more and more attention in recent years, for making full use of image global correlation to improve performance on various computer vision applications. However, very few studies focus on solving CSC based image Super-Resolution (SR) problem. As a consequence, there is no significant progress in this area over a period of time. In this paper, we exploit the natural connection between CSC and Convolutional Neural Networks (CNN) to address CSC based image SR. Specifically, Convolutional Iterative Soft Thresholding Algorithm (CISTA) is introduced to solve CSC problem and it can be implemented using CNN architectures. Then we develop a novel CSC based SR framework analogy to the traditional SC based SR methods. Two models inspired by this framework are proposed for pre-/post-upsampling SR, respectively. Compared with recent state-of-the-art SR methods, both of our proposed models show superior performance in terms of both quantitative and qualitative measurements.

## 1 Introduction

Single Image Super-Resolution (SISR), which aims to restore a visually pleasing High-Resolution (HR) image from its Low-Resolution (LR) version, is still a challenging task within computer vision research community (Timofte et al., 2017; 2018). Since multiple solutions exist for the mapping from LR to HR space, SISR is highly ill-posed. To regularize the solution of SISR, various priors of natural images have been exploited, especially the current leading learning-based methods (Wang et al., 2015; Dong et al., 2016; Mao et al., 2016; Kim et al., 2016a;b; Tai et al., 2017a;b; Lim et al., 2017; Ahn et al., 2018; Haris et al., 2018; Li et al., 2018; Zhang et al., 2018) are proposed to directly learn the non-linear LR-HR mapping.

By modeling the sparse prior in natural images, the Sparse Coding (SC) based methods for SR (Yang et al., 2008; 2010; 2014) with strong theoretical support are widely used owing to their excellent performance. Considering the complexity in images, these methods divide the image into overlapping patches and aim to jointly train two over-complete dictionaries for LR/HR patches. There are usually three steps in these methods' framework. First, overlapping patches are extracted from input image. Then to reconstruct the HR patch, the sparse representation of LR patch can be applied to the HR dictionary with *the assumption that LR/HR patch pair shares similar sparse representation*. The final HR image is produced by aggregating the recovered HR patches.

Recently, with the development of Deep Learning (DL), many researchers attempt to combine the advantages of DL and SC for image SR. Dong et al. (Dong et al., 2016) firstly proposed the seminal CNN model for SR termed as SRCNN, which exploits a shallow convolutional neural network to learn a nonlinear LR-HR mapping in an end-to-end manner and dramatically overshadows conventional methods (Yang et al., 2010; Timofte et al., 2014). However, sparse prior is ignored to a large extent in SRCNN for it adopts a generic architecture without considering the domain expertise. To address this issue, Wang et al. (Wang et al., 2015) implemented a Sparse Coding based Network (SCN) for image SR, by combining the merits of sparse coding and deep learning, which fully exploits the approximation of sparse coding learned from the LISTA (Gregor & LeCun, 2010) based sub-network.

It's worth to note that most of SC based methods utilize the sparse prior locally (Papyan et al., 2017b), i.e., coping with overlapping image patches. Thus the consistency of pixels in overlapped

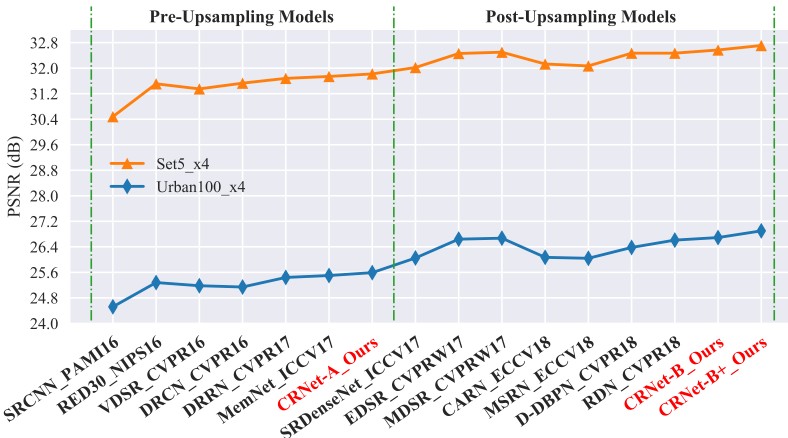

Figure 1: PSNRs of recent state-of-the-arts for scale factor ×4 on Set5 (Bevilacqua et al., 2012) and Urban100 (Huang et al., 2015). Red names represent our proposed models.

patches has been ignored (Gu et al., 2015; Papyan et al., 2017b). To address this issue, CSC is proposed to serve sparse prior as a global prior (Zeiler et al., 2010; Papyan et al., 2017b; 2018) and it furnishes a way to fill the local-global gap by working directly on the entire image by convolution operation. Consequently, CSC has attained much attention from researchers (Zeiler et al., 2010; Bristow et al., 2013; Heide et al., 2015; Gu et al., 2015; Sreter & Giryes, 2018; Garcia-Cardona & Wohlberg, 2018a). However, very few studies focus on the validation of CSC for image SR (Gu et al., 2015), resulting in no work been reported that CSC based image SR can achieve state-of-the-art performance. Can CSC based image SR show highly competitive results with recent state-of-the-art methods (Dong et al., 2016; Mao et al., 2016; Kim et al., 2016a;b; Tai et al., 2017a;b; Tong et al., 2017; Lim et al., 2017; Li et al., 2018; Zhang et al., 2018)? To answer this question, the following issues need to be considered:

**Framework Issue**. Compared with SC based image SR methods (Yang et al., 2008; 2010), the lack of a unified framework has hindered progress towards improving the performance of CSC based image SR.

**Optimization Issue**. The previous CSC based image SR method (Gu et al., 2015) contains several steps and they are optimized independently. Hundreds of iterations are required to solve the CSC problem in each step.

**Memory Issue**. To solve the CSC problem, ADMM (Boyd et al., 2011) is commonly employed (Bristow et al., 2013; Wohlberg, 2014; Heide et al., 2015; Wohlberg, 2016; Garcia-Cardona & Wohlberg, 2018a), where the whole training set needs to be loaded in memory. As a consequence, it is not applicable to improve the performance by enlarging the training set.

**Multi-Scale Issue**. Training a single model for multiple scales is difficult for the previous CSC based image SR method (Gu et al., 2015).

Based on these considerations, in this paper, we attempt to answer the aforementioned question. Specifically, we exploit the advantages of CSC and the powerful learning ability of deep learning to address image SR problem. Moreover, massive theoretical foundations for CSC (Papyan et al., 2017b; 2018; Garcia-Cardona & Wohlberg, 2018b) make our proposed architectures interpretable and also enable to theoretically analyze our SR performance. In the rest of this paper, we first introduce CISTA, which can be naturally implemented using CNN architectures for solving the CSC problem. Then we develop a framework for CSC based image SR, which can address the **Framework Issue**. Subsequently, CRNet-A (CSC and Residual learning based Network) and CRNet-B inspired by this framework are proposed for image SR. They are classified as pre- and post-upsampling models (Wang et al., 2019) respectively, as the former takes Interpolated LR (ILR) images as input while the latter processes LR images directly. By adopting CNN architectures, **Optimization Issue** and **Memory Issue** would be mitigated to some extent. For **Multi-Scale Issue**, with the help of the recently introduced scale augmentation (Kim et al., 2016a;b) or scale-specific multi-path learning (Lim et al., 2017; Wang et al., 2019) strategies, both of our models are capable of handling multi-

scale SR problem effectively, and achieve favorable performance against state-of-the-arts, as shown in Fig. 1. The main contributions of this paper include:

- We introduce CISTA, which can be naturally implemented using CNN architectures for solving the CSC problem.
- A novel framework for CSC based image SR is developed. Two models, CRNet-A and CRNet-B, inspired by this framework are proposed for image SR.
- The differences between our proposed models and several SR models with recursive learning strategy, e.g., DRRN (Tai et al., 2017a), SCN (Wang et al., 2015), DRCN (Kim et al., 2016b), are discussed.

## 2 RELATED WORK

Sparse coding has been widely used in a variety of applications (Zhang et al., 2015). As for SISR, Yang et al. (Yang et al., 2008) proposed a representative Sparse coding based Super-Resolution (ScSR) method. In the training stage, ScSR attempts to learn the LR/HR overcomplete dictionary pair $\boldsymbol{D}_l/\boldsymbol{D}_h$ jointly by given a group of LR/HR training patch pairs $\boldsymbol{x}_l/\boldsymbol{x}_h$. In the test stage, the HR patch $\boldsymbol{x}_h$ is reconstructed from its LR version $\boldsymbol{x}_l$ by assuming they share the same sparse code. Specifically, the optimal sparse code is obainted by minimizing the following sparsity-inducing $\ell_1$-norm regularized objective function

$$\boldsymbol{z}^* = \arg\min_{\boldsymbol{z}} \|\boldsymbol{x}_l - \boldsymbol{D}_l\boldsymbol{z}\|_2^2 + \lambda\|\boldsymbol{z}\|_1, \tag{1}$$

then the HR patch is obtained by $\boldsymbol{x}_h = \boldsymbol{D}_h\boldsymbol{z}^*$. Finally, the HR image can be estimated by aggregating all the reconstructed HR patches. Inspired by ScSR, many SC based methods have been proposed with various constraints on sparse code or dictionary Yang et al. (2012); Wang et al. (2012).

Traditional SC based SR algorithms usually process images in a patch based manner to reduce the burden of modeling and computation, resulting in the inconsistency problem (Papyan et al., 2017b). As a special case of SC, CSC is inherently suitable (Zeiler et al., 2010) and proposed to avoid the inconsistency problem by representing the whole image directly. Specifically, an image $\boldsymbol{y} \in \mathbb{R}^{n_r \times n_c}$ can be represented as the summation of $m$ feature maps $\boldsymbol{z}_i \in \mathbb{R}^{n_r \times n_c}$ convolved with the corresponding filters $\boldsymbol{f}_i \in \mathbb{R}^{s \times s}$: $\boldsymbol{y} = \sum_{i=1}^m \boldsymbol{f}_i \otimes \boldsymbol{z}_i$, where $\otimes$ is the convolution operation.

Gu et al. (Gu et al., 2015) proposed the CSC-SR method and revealed the potential of CSC for image SR. In (Gu et al., 2015), CSC-SR requires to solve the following CSC based optimization problem in both the training and testing phase:

$$\min_{\boldsymbol{f},\boldsymbol{z}} \frac{1}{2}\left\|\boldsymbol{y} - \sum_{i=1}^m \boldsymbol{f}_i \otimes \boldsymbol{z}_i\right\|_2^2 + \lambda\sum_{i=1}^m \|\boldsymbol{z}_i\|_1. \tag{2}$$

which is solved by alternatively optimizing the $\boldsymbol{z}$ and $\boldsymbol{f}$ subproblems (Wohlberg, 2014). The $\boldsymbol{z}$ subproblem is a standard CSC problem. Hundreds of iterations are required to solve the CSC problem and the aforementioned **Optimization Issue** and **Memory Issue** cannot be completely avoided. Inspired by the success of deep learning based sparse coding (Gregor & LeCun, 2010), we exploit the natural connection between CSC and CNN to solve the CSC problem efficiently.

## 3 CISTA FOR SOLVING CSC PROBLEM

CSC can be considered as a special case of conventional SC, due to the fact that convolution operation can be replaced with matrix multiplication, so the objective function of CSC can be formulated as:

$$\min_{\boldsymbol{z}} \left\|\boldsymbol{y} - \sum_{i=1}^m \boldsymbol{F}_i\boldsymbol{z}_i\right\|_2^2 + \lambda\sum_{i=1}^m \|\boldsymbol{z}_i\|_1. \tag{3}$$

$\boldsymbol{y}, \boldsymbol{z}_i$ are in vectorized form and $\boldsymbol{F}_i$ is a sparse convolution matrix with the following attributes:

$$\begin{aligned} \boldsymbol{F}_i\boldsymbol{z}_i &\equiv \boldsymbol{f}_i \otimes \boldsymbol{z}_i \\ \boldsymbol{F}_i^T\boldsymbol{z}_i &\equiv \texttt{flipud}(\texttt{fliplr}(\boldsymbol{f}_i)) \otimes \boldsymbol{z}_i \\ &\equiv \texttt{flip}(\boldsymbol{f}_i) \otimes \boldsymbol{z}_i \end{aligned} \tag{4}$$

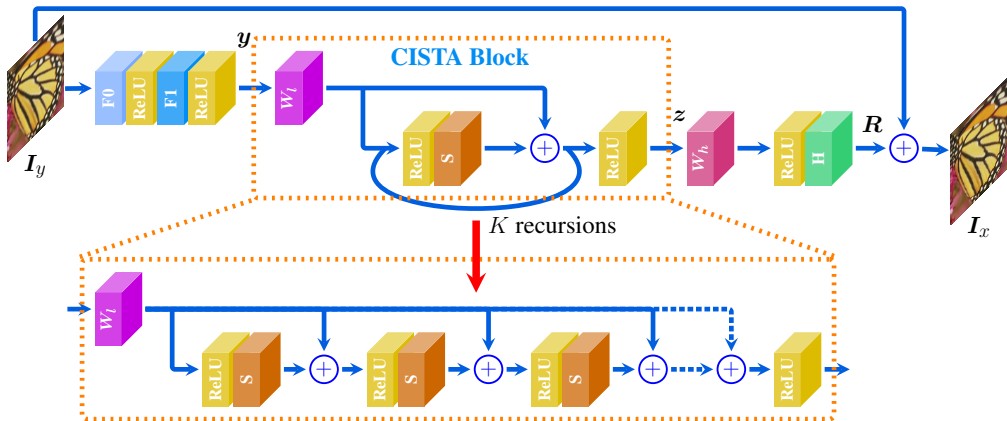

Figure 2: The architecture of the pre-upsampling model CRNet-A. The proposed CISTA block with $K$ recursions is surrounded by the dashed box and its unfolded version is shown in the bottom. $\boldsymbol{S}$ is shared across every recursion.

where $\texttt{fliplr}(\cdot)$ and $\texttt{flipud}(\cdot)$ are following the notations of Zeiler et al.(Zeiler et al., 2010), representing that array is flipped in left/right or up/down direction.

Iterative Soft Thresholding Algorithm (ISTA) (Daubechies et al., 2004) can be utilized to solve (3), at the $k^{th}$ iteration:

$$z_{k+1} = h_\theta \left( z_k + \frac{1}{L} \boldsymbol{F}^T \left( \boldsymbol{y} - \boldsymbol{F} z_k \right) \right) \tag{5}$$

where $L$ is the Lipschitz constant, $\boldsymbol{F} = [\boldsymbol{F}_1, \boldsymbol{F}_2, \ldots, \boldsymbol{F}_m]$ and $\boldsymbol{F}\boldsymbol{z} = \sum_{i=1}^m \boldsymbol{F}_i \boldsymbol{z}_i$. Using the relation in (4) to replace the matrix multiplication with convolution operator, we can reformulate (5) as:

$$z_{k+1} = h_\theta \left( \boldsymbol{I} z_k + \frac{1}{L} \texttt{flip}(\boldsymbol{f}) \otimes \left( \boldsymbol{y} - \boldsymbol{f} \otimes z_k \right) \right) \tag{6}$$

where $\boldsymbol{f} = [\boldsymbol{f}_1, \boldsymbol{f}_2, \ldots, \boldsymbol{f}_m]$, $\texttt{flip}(\boldsymbol{f}) = [\texttt{flip}(\boldsymbol{f}_1), \texttt{flip}(\boldsymbol{f}_2), \ldots, \texttt{flip}(\boldsymbol{f}_m)]$ and $\boldsymbol{I}$ is the identity matrix. Note that identity matrix $\boldsymbol{I}$ is also a sparse convolution matrix, so according to (4), there existing a filter $\boldsymbol{n}$ satisfies:

$$\boldsymbol{I}\boldsymbol{z} = \boldsymbol{n} \otimes \boldsymbol{z}, \tag{7}$$

so (6) becomes:

$$z_{k+1} = h_\theta \left( \boldsymbol{W} \otimes \boldsymbol{y} + \boldsymbol{S} \otimes z_k \right), \tag{8}$$

where $\boldsymbol{W} = \frac{1}{L} \texttt{flip}(\boldsymbol{f})$ and $\boldsymbol{S} = \boldsymbol{n} - \frac{1}{L} \texttt{flip}(\boldsymbol{f}) \otimes \boldsymbol{f}$. Even though (3) is for a single image with one channel, the extension to multiple channels (for both image and filters) and multiple images is mathematically straightforward. Thus for $\boldsymbol{y} \in \mathbb{R}^{b \times c \times n_r \times n_c}$ representing $b$ images of size $n_r \times n_c$ with $c$ channels, (8) is still true with $\boldsymbol{W} \in \mathbb{R}^{m \times c \times s \times s}$ and $\boldsymbol{S} \in \mathbb{R}^{m \times m \times s \times s}$.

As for $h_{\boldsymbol{\theta}}$, (Papyan et al., 2017a) reveals two important facts: (1) the expressiveness of the sparsity inspired model is not affected even by restricting the coefficients to be nonnegative; (2) the $ReLU$ (Nair & Hinton, 2010) activation function and the soft nonnegative thresholding operator are equal, that is:

$$h_{\boldsymbol{\theta}}^+ (\boldsymbol{\alpha}) = \max(\boldsymbol{\alpha} - \boldsymbol{\theta}, 0) = ReLU(\boldsymbol{\alpha} - \boldsymbol{\theta}). \tag{9}$$

We set $\boldsymbol{\theta} = \boldsymbol{0}$ for simplicity. So the final form of (8) is:

$$z_{k+1} = ReLU \left( \boldsymbol{W} \otimes \boldsymbol{y} + \boldsymbol{S} \otimes z_k \right). \tag{10}$$

One can see that (10) is a convolutional form of (5), so we name it as CISTA. It provides the solution of (3) with theoretical guarantees (Daubechies et al., 2004). Furthermore, this convolutional form can be implemented employing CNN architectures. So $\boldsymbol{W}$ and $\boldsymbol{S}$ in (10) would be trainable.

## 4 PROPOSED METHOD

In this work, exploiting the basic CISTA block, we implement the SR using CNN techniques. With more recursions used in CISTA, the network becomes deeper and tends to be bothered by the gradient vanishing/exploding problems. Residual learning (Kim et al., 2016a;b; Tai et al., 2017a) is such a useful tool that not only mitigates these difficulties, but helps network converge faster (A.1).

### 4.1 CRNET-A MODEL FOR PRE-UPSAMPLING

As shown in Fig. 2, CRNet-A takes the ILR image $I_y$ with $c$ channels as input, and predicts the output HR image as $I_x$. Two convolution layers, $F_0 \in \mathbb{R}^{n_0 \times c \times s \times s}$ consisting of $n_0$ filters of spatial size $c \times s \times s$ and $F_1 \in \mathbb{R}^{n_0 \times n_0 \times s \times s}$ containing $n_0$ filters of spatial size $n_0 \times s \times s$ are utilized for hierarchical features extraction from ILR image:

$$y = ReLU\Big(F_1 \otimes ReLU(F_0 \otimes I_y)\Big). \tag{11}$$

The ILR features are then fed into a CISTA block to learn the convolutional sparse codes. As stated in (10), two convolutional layers $W_l \in \mathbb{R}^{m_0 \times n_0 \times s \times s}$ and $S \in \mathbb{R}^{m_0 \times m_0 \times s \times s}$ are needed:

$$z_{k+1} = ReLU(W_l \otimes y + S \otimes z_k), \tag{12}$$

where $z_0$ is initialized to $ReLU(W_l \otimes y)$. The convolutional sparse codes $z$ are learned after $K$ recursions with $S$ shared across every recursion. When the convolutional sparse codes $z$ are obtained, it is then passed through a convolution layer $W_h \in \mathbb{R}^{n_0 \times m_0 \times s \times s}$ to recover the HR feature maps. The last convolution layer $H \in \mathbb{R}^{c \times n_0 \times s \times s}$ is used as HR filters:

$$R = H \otimes ReLU(W_h \otimes z). \tag{13}$$

Note that we pad zeros before all convolution operations to keep all the feature maps to have the same size, which is a common strategy used in a variety of methods (Kim et al., 2016a;b; Tai et al., 2017a). So the residual image $R$ has the same size as the input ILR image $I_y$, and the final HR image $I_x$ would be reconstructed by:

$$I_x = I_y + R. \tag{14}$$

### 4.2 CRNET-B MODEL FOR POST-UPSAMPLING

We extend CRNet-A to its post-upsampling version to further mine its potential. Notice that most post-upsampling models (Ledig et al., 2017; Tong et al., 2017; Li et al., 2018; Zhang et al., 2018) need to train and store many scale-dependent models for various scales without fully using the inter-scale correlation, so we adopt the scale-specific multi-path learning strategy (Wang et al., 2019) presented in MDSR (Lim et al., 2017) with minor modifications to address this issue. The complete model is shown in A.2. The main branch is our CRNet-A module. The pre-processing modules are used for reducing the variance from input images of different scales and only one residual unit with $3 \times 3$ kernels is used in each of the pre-processing module. At the end of CRNet-B, upsampling modules are used for multi-scale reconstruction.

## 5 EXPERIMENTAL RESULTS

### 5.1 SETTINGS

**Datasets and Metrics** Being fair to (Kim et al., 2016a; Tai et al., 2017a; Zhang et al., 2018), 291(Yang et al., 2010; Martin et al., 2001) images are used for CRNet-A, while 800 (Timofte et al., 2017) images for CRNet-B. During testing, *Set5* (Bevilacqua et al., 2012), *Set14* (Zeyde et al., 2010), *B100* (Martin et al., 2001), *Urban100* (Huang et al., 2015) and *Manga109* (Matsui et al., 2017) are employed. Both PSNR and SSIM (Wang et al., 2004) on Y channel of transformed YCbCr space are calculated for evaluation.

**Parameter Settings** We set $n_0 = 128$, $m_0 = 256$ for CRNet-A (*i.e.* every convolution layer contains 128 filters while $W_l$ and $S$ have 256 filters), while $n_0 = 64$, $m_0 = 1024$ for CRNet-B. We choose $K = 25$ in both of our models. We implement our models using the PyTorch (Paszke et al., 2017) framework with NVIDIA Titan Xp. More training details and parameter study could be found in A.4 and A.5.

Table 2: Average PSNR/SSIMs of **Pre-upsampling** models for scale factor $\times 2$, $\times 3$ and $\times 4$ on datasets Set5, Set14, BSD100 and Urban100. Red: the best; blue: the second best.

| Dataset | Scale | Bicubic | VDSR (Kim et al., 2016a) | DRCN (Kim et al., 2016b) | DRRN (Tai et al., 2017a) | MemNet (Tai et al., 2017b) | CRNet-A (ours) |
|---|---|---|---|---|---|---|---|
| Set5 | $\times 2$ | 33.66/0.9299 | 37.53/0.9587 | 37.63/0.9588 | 37.74/0.9591 | 37.78/0.9597 | 37.79/0.9600 |
| | $\times 3$ | 30.39/0.8682 | 33.66/0.9213 | 33.82/0.9226 | 34.03/0.9244 | 34.09/0.9248 | 34.11/0.9254 |
| | $\times 4$ | 28.42/0.8104 | 31.35/0.8838 | 31.53/0.8854 | 31.68/0.8888 | 31.74/0.8893 | 31.82/0.8907 |
| Set14 | $\times 2$ | 30.24/0.8688 | 33.03/0.9124 | 33.04/0.9118 | 33.23/0.9136 | 33.28/0.9142 | 33.33/0.9152 |
| | $\times 3$ | 27.55/0.7742 | 29.77/0.8314 | 29.76/0.8311 | 29.96/0.8349 | 30.00/0.8350 | 29.99/0.8359 |
| | $\times 4$ | 26.00/0.7027 | 28.01/0.7674 | 28.02/0.7670 | 28.21/0.7720 | 28.26/0.7723 | 28.29/0.7741 |
| B100 | $\times 2$ | 29.56/0.8431 | 31.90/0.8960 | 31.85/0.8942 | 32.05/0.8973 | 32.08/0.8978 | 32.09/0.8985 |
| | $\times 3$ | 27.21/0.7385 | 28.82/0.7976 | 28.80/0.7963 | 28.95/0.8004 | 28.96/0.8001 | 28.99/0.8021 |
| | $\times 4$ | 25.96/0.6675 | 27.29/0.7251 | 27.23/0.7233 | 27.38/0.7284 | 27.40/0.7281 | 27.44/0.7302 |
| Urban100 | $\times 2$ | 26.88/0.8403 | 30.76/0.9140 | 30.75/0.9133 | 31.23/0.9188 | 31.31/0.9195 | 31.36/0.9207 |
| | $\times 3$ | 24.46/0.7349 | 27.14/0.8279 | 27.15/0.8276 | 27.53/0.8378 | 27.56/0.8376 | 27.64/0.8403 |
| | $\times 4$ | 23.14/0.6577 | 25.18/0.7524 | 25.14/0.7510 | 25.44/0.7638 | 25.50/0.7630 | 25.59/0.7680 |

## 5.2 COMPARISON WITH CSC-SR

We first compare our proposed models with the existing CSC based image SR method, i.e., CSC-SR (Gu et al., 2015). Since CSC-SR utilizes LR images as input image, it can be considered as a post-upsampling method, thus CRNet-B is used for comparison. Tab. 1 presents that our CRNet-B clearly outperforms CSC-SR by a large margin.

Table 1: **Left Table:** Average PSNR/SSIMs of CSC-SR (Gu et al., 2015) and CRNet-B for scale factor $\times 2$, $\times 3$ and $\times 4$ on Set5, Set14 and B100. Red: the best; blue: the second best. **Right Table:** Ablation Study of LRL, PA and CC.

| Dataset | Scale | Bicubic | CSC-SR | CRNet-B |
|---|---|---|---|---|
| Set5 | $\times 2$ | 33.66 / 0.9299 | 36.62 / 0.9549 | 38.13 / 0.9610 |
| | $\times 3$ | 30.39 / 0.8682 | 32.65 / 0.9098 | 34.75 / 0.9296 |
| | $\times 4$ | 28.42 / 0.8104 | 30.36 / 0.8607 | 32.57 / 0.8991 |
| Set14 | $\times 2$ | 30.24 / 0.8688 | 32.30 / 0.9070 | 34.09 / 0.9219 |
| | $\times 3$ | 27.55 / 0.7742 | 29.14 / 0.8208 | 30.58 / 0.8465 |
| | $\times 4$ | 26.00 / 0.7027 | 27.30 / 0.7499 | 28.79 / 0.7867 |
| B100 | $\times 2$ | 29.56 / 0.8431 | 31.27 / 0.8876 | 32.32 / 0.9014 |
| | $\times 3$ | 27.21 / 0.7385 | 28.31 / 0.7857 | 29.26 / 0.8091 |
| | $\times 4$ | 25.96/0.6675 | 26.82 / 0.7101 | 27.73 / 0.7414 |

| LRL | PA | CC | PSNR |
|---|---|---|---|
| ✗ | ✗ | ✗ | 33.53 |
| ✔ | ✗ | ✗ | 33.54 |
| ✗ | ✔ | ✗ | 33.52 |
| ✗ | ✗ | ✔ | 34.05 |
| ✔ | ✔ | ✗ | 33.58 |
| ✔ | ✗ | ✔ | 34.09 |
| ✗ | ✔ | ✔ | 33.97 |
| ✔ | ✔ | ✔ | **34.15** |

## 5.3 COMPARISON WITH STATE OF THE ARTS

We now compare the proposed models with other state-of-the-arts in recent years. We compare CRNet-A with pre-upsampling models (i.e., VDSR (Kim et al., 2016a), DRCN (Kim et al., 2016b), DRRN (Tai et al., 2017a), MemNet (Tai et al., 2017b)) while CRNet-B with post-upsampling architectures (i.e., SRDenseNet (Tong et al., 2017), MSRN (Li et al., 2018), D-DBPN (Haris et al., 2018), EDSR (Lim et al., 2017), RDN (Zhang et al., 2018)). Similar to (Lim et al., 2017; Zhang et al., 2018), self-ensemble strategy (Lim et al., 2017) is also adopted to further improve the performance of CRNet-B, and we denote the self-ensembled version as CRNet-B+.

Tab. 2 and Tab. 3 show the quantitative comparisons on the benchmark testing sets. Both of our models achieve superior performance against the state-of-the-arts, which indicates the effectiveness of our models. Qualitative results are provided in Fig. 3. Our methods tend to produce shaper edges and more correct textures, while other images may be blurred or distorted. More visual comparisons are available in A.6.

Fig. 4(a) and 4(b) shows the performance versus the number of parameters, our CRNet-B and CRNet-B+ achieve better results with fewer parameters than EDSR (Lim et al., 2017) and RDN (Zhang et al., 2018). It's worth noting that EDSR/MDSR and RDN are far deeper than CRNet-B (e.g., 169 vs. 36), but CRNet-B is quite wider ($W_l$ and $S$ have $1,024$ filters). As reported in (Lim et al., 2017), when increasing the number of filters to a certain level, e.g., 256, the training procedure of EDSR (for $\times 2$) without residual scaling (Szegedy et al., 2018; Lim et al., 2017) is numerically unstable, as shown in Fig. 4(c). However, CRNet-B is relieved from the residual scaling

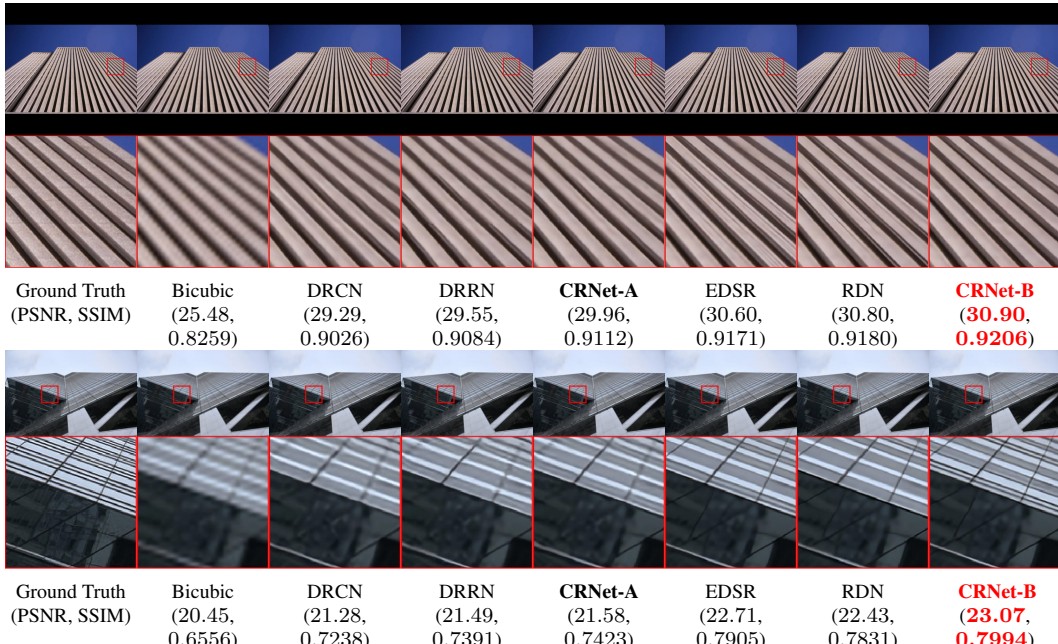

| Ground Truth (PSNR, SSIM) | Bicubic (25.48, 0.8259) | DRCN (29.29, 0.9026) | DRRN (29.55, 0.9084) | **CRNet-A** (29.96, 0.9112) | EDSR (30.60, 0.9171) | RDN (30.80, 0.9180) | **CRNet-B** (**30.90**, **0.9206**) |
|---|---|---|---|---|---|---|---|

| Ground Truth (PSNR, SSIM) | Bicubic (20.45, 0.6556) | DRCN (21.28, 0.7238) | DRRN (21.49, 0.7391) | **CRNet-A** (21.58, 0.7423) | EDSR (22.71, 0.7905) | RDN (22.43, 0.7831) | **CRNet-B** (**23.07**, **0.7994**) |
|---|---|---|---|---|---|---|---|

Figure 3: SR results of "img016" and "img059" from **Urban100** with scale factor ×4. Red indicates the best performance.

Table 3: Average PSNR/SSIMs of **Post-upsampling** models for scale factor ×2, ×3 and ×4 on datasets Set5, Set14, BSD100, Urban100 and Manga109. Red: the best; blue: the second best.

| Dataset | Scale | MSRN Li et al. (2018) | D-DBPN Haris et al. (2018) | EDSR (Lim et al., 2017) | RDN (Zhang et al., 2018) | CRNet-B (ours) | CRNet-B+ (ours) |
|---|---|---|---|---|---|---|---|
| Set5 | ×2 | 38.08/0.9605 | 38.09/0.9600 | 38.11/0.9601 | 38.24/0.9614 | 38.13/0.9610 | 38.25/0.9614 |
| | ×3 | 34.38/0.9262 | -/- | 34.65/0.9282 | 34.71/0.9296 | 34.75/0.9296 | 34.83/0.9303 |
| | ×4 | 32.07/0.8903 | 32.47/0.8980 | 32.46/0.8968 | 32.47/0.8990 | 32.57/0.8991 | 32.71/0.9008 |
| Set14 | ×2 | 33.74/0.9170 | 33.85/0.9190 | 33.92/0.9195 | 34.01/0.9212 | 34.09/0.9219 | 34.15/0.9227 |
| | ×3 | 30.34/0.8395 | -/- | 30.52/0.8462 | 30.57/0.8468 | 30.58/0.8465 | 30.67/0.8481 |
| | ×4 | 28.60/0.7751 | 28.82/0.7860 | 28.80/0.7876 | 28.81/0.7871 | 28.79/0.7867 | 28.93/0.7894 |
| B100 | ×2 | 32.23/0.9013 | 32.27/0.9000 | 32.32/0.9013 | 32.34/0.9017 | 32.32/0.9014 | 32.38/0.9020 |
| | ×3 | 29.08/0.8041 | -/- | 29.25/0.8093 | 29.26/0.8093 | 29.26/0.8091 | 29.32/0.8103 |
| | ×4 | 27.52/0.7273 | 27.72/0.7400 | 27.71/0.7420 | 27.72/0.7419 | 27.73/0.7414 | 27.80/0.7430 |
| Urban100 | ×2 | 32.22/0.9326 | 32.55/0.9324 | 32.93/0.9351 | 32.89/0.9353 | 32.93/0.9355 | 33.14/0.9370 |
| | ×3 | 28.08/0.8554 | -/- | 28.80/0.8653 | 28.80/0.8653 | 28.87/0.8667 | 29.09/0.8697 |
| | ×4 | 26.04/0.7896 | 26.38/0.7946 | 26.64/0.8033 | 26.61/0.8028 | 26.69/0.8045 | 26.90/0.8089 |
| Manga109 | ×2 | 38.82/0.9868 | 38.89/0.9775 | 39.10/0.9773 | 39.18/0.9780 | 39.07/0.9778 | 39.28/0.9784 |
| | ×3 | 33.44/0.9427 | -/- | 34.17/0.9476 | 34.13/0.9484 | 34.17/0.9481 | 34.52/0.9498 |
| | ×4 | 30.17/0.9034 | 30.91/0.9137 | 31.02/0.9148 | 31.00/0.9151 | 31.16/0.9154 | 31.52/0.9187 |

trick. The training loss of CRNet-B is depicted in Fig. 4(d), it converges fast at the begining, then keeps decreasing and finally fluctuates at a certain range.

## 5.4 ABLATION STUDY

The proposed networks combine the merits of CSC and Residual learning. Particularly, CSC problem is solved in a learnable manner, namely CISTA, where three techniques are naturally inferred, i.e. **Local Residual Learning (LRL)**, **Pre-Activation (PA)** and **Consistency Constraint (CC)**. **Ablation studies** are implemented to address the superiority of the proposed networks with the elementary blocks of CISTA by combining LRL, PA and CC. It is shown in Tab. 1 that the network simultaneously containing LRL, PA and CC gives the best performance.

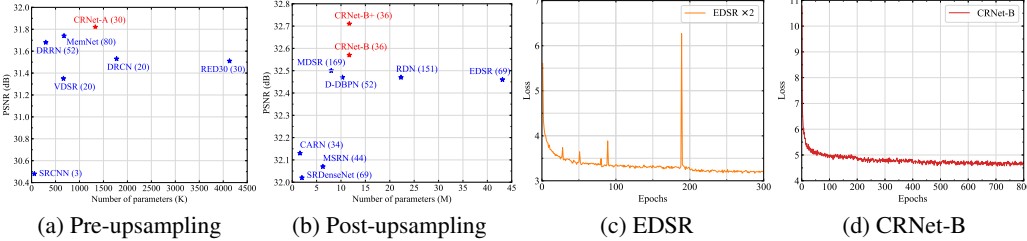

| (a) Pre-upsampling | (b) Post-upsampling | (c) EDSR | (d) CRNet-B |

Figure 4: **(a) and (b)**: PSNR of recent state-of-the-arts versus the number of parameters for scale factor ×4 on Set5. The number of layers are marked in the parentheses; **(c) and (d)**: Training loss of EDSR without residual scaling and CRNet-B.

## 6 DISCUSSIONS

We discuss the differences between our proposed models and several recent CNN models for SR *with recursive learning strategy*, i.e., DRRN (Tai et al., 2017a), SCN (Wang et al., 2015) and DRCN (Kim et al., 2016b). Due to the fact that CRNet-B is an extension of CRNet-A, i.e., the main part of CRNet-B has the same structure as CRNet-A, so we use CRNet-A here for comparison. The simplified structures of these models are shown in A.3, where the digits on the left of the recursion line represent the number of recursions.

**Difference to DRRN**. The main part of DRRN (Tai et al., 2017a) is the recursive block structure, where several residual units with BN layers are stacked. On the other hand, guided by (10), CRNet-A contains no BN layers. Coinciding with EDSR/MDSR (Lim et al., 2017), by normalizing features, BN layers get rid of range flexibility from networks. Furthermore, BN consumes much amount of GPU memory and increases computational complexity. Experimental results on benchmark datasets under common-used assessments demonstrate the superiority of CRNet-A.

**Difference to SCN**. There are two main differences between CRNet-A and SCN (Wang et al., 2015): CISTA block and residual learning. Specifically, CRNet-A takes consistency constraint into consideration with the help of CISTA block, while SCN uses linear layers and ignores the information from the consistency prior. On the other hand, CRNet-A adopts residual learning, which is a powerful tool for training deeper networks. CRNet-A (30 layers) is much deeper than SCN (5 layers). As indicated in (Kim et al., 2016a), a deeper network has larger receptive fileds, so more contextual information in an image would be utilized to infer high-frequency details. In A.1, we show that more recursions, e.g., 48, can be used to achieve better performance.

**Difference to DRCN**. CRNet-A differs with DRCN (Kim et al., 2016b) in two aspects: recursive block and training techniques. In the recursive block, both local residual learning (Tai et al., 2017a) and pre-activation (He et al., 2016; Tai et al., 2017a) are utilized in CRNet-A, which are demonstrated to be effective in (Tai et al., 2017a). As for training techniques, DRCN is not easy to train, so recursive-supervision is introduced to facilitate the network to converge. Moreover, an ensemble strategy (the final output is the weighted average of all intermediate predictions) is used to further improve the performance. CRNet-A is relieved from these techniques and can be easily trained with more recursions.

## 7 CONCLUSIONS

In this work, we propose two effective CSC based image SR models, i.e., CRNet-A and CRNet-B, for pre-/post-upsampling SR, respectively. By combining the merits of CSC and CNN, we achieve superior performance against recent state-of-the-arts. Furthermore, our framework and CISTA block are expected to be applicable in various CSC based tasks, though in this paper we focus on CSC based image SR.

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

# A APPENDIX

## A.1 RESIDUAL *vs.* NON-RESIDUAL

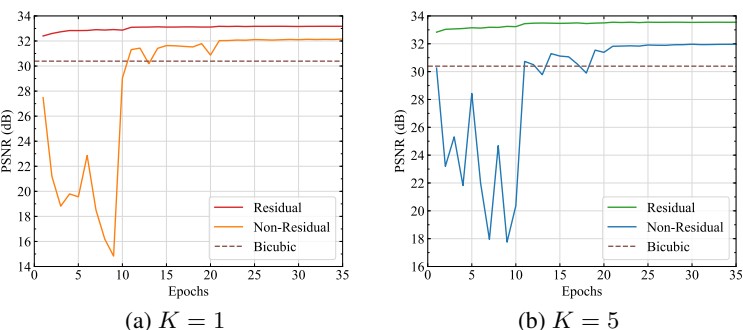

(a) $K = 1$      (b) $K = 5$

Figure 5: Performance curve for residual/non-residual networks with different recursions. The tests are conducted on Set5 for scale factor $\times 3$.

## A.2 CRNET-B

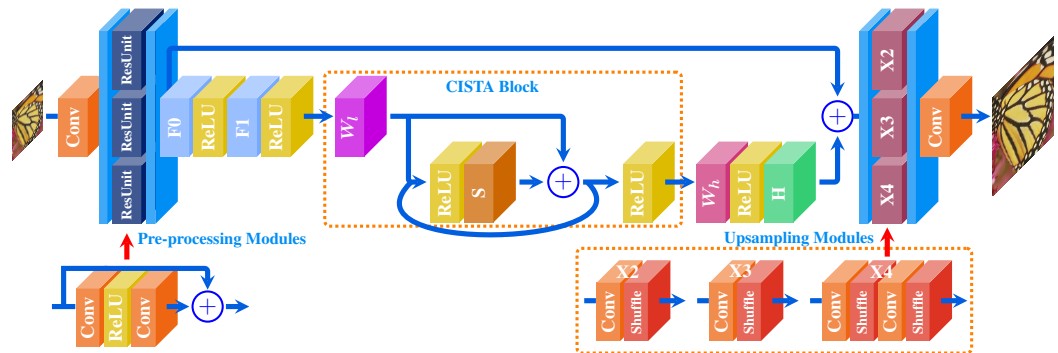

Figure 6: The architecture of the post-upsampling model CRNet-B.

## A.3 GRAPHICAL COMPARISON TO OTHER RECURRENT CNNS IN SR

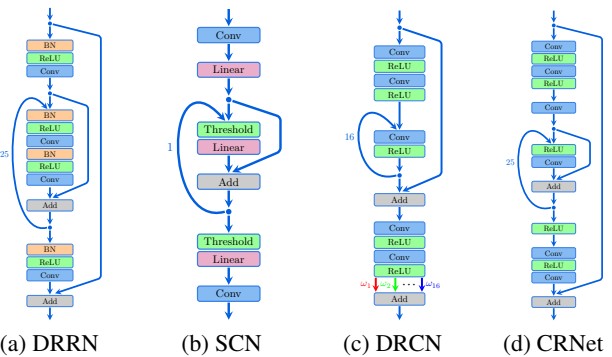

(a) DRRN     (b) SCN     (c) DRCN     (d) CRNet

Figure 7: Simplified network structures of (a) DRRN (Tai et al., 2017a), (b) SCN (Wang et al., 2015), (c) DRCN (Kim et al., 2016b), (d) our model CRNet.

## A.4 IMPLEMENTATION DETAILS

**CRNet-A** To enlarge the training set, data augmentation, which includes flipping (horizontally and vertically), rotating (90, 180, and 270 degrees), scaling (0.7, 0.5 and 0.4), is performed on each image of *291* dataset, where 91 of these images are from Yang et al. (Yang et al., 2010) with the

addition of 200 images from Berkeley Segmentation Dataset (Martin et al., 2001). Scale augmentation is exploited by combining images of different scales ($\times 2$, $\times 3$ and $\times 4$) into one training set. Furthermore, all training images are partitioned into $33 \times 33$ patches with the stride of 33, providing a total of $1,929,728$ ILR-HR training pairs. Every convolution layer in CRNet-A contains 128 filters ($n_0 = 128$) of size $3 \times 3$ while $\boldsymbol{W}_l$ and $\boldsymbol{S}$ have 256 filters ($m_0 = 256$). We follow the same strategy as He et al. (He et al., 2015) for weight initialization where all weights are drawn from a normal distribution with zero mean and variance $2/n_{out}$, where $n_{out}$ is the number of output units. The network is optimized using SGD with mini-batch size of 128, momentum parameter of 0.9 and weight decay of $10^{-4}$. The learning rate is initially set to 0.1 and then decreased by a factor of 10 every 10 epochs. We train a total of 35 epochs as no further decrease of the loss can be observed. For maximal convergence speed, we utilize the adjustable gradient clipping strategy stated in (Kim et al., 2016a), with gradients clipped to $[-\theta, \theta]$, where $\theta = 0.4$ is the gradient clipping parameter.

**CRNet-B** We use the 800 training images of the DIV2K (Timofte et al., 2017) dataset to train CRNet-B, and all the images are pre-processed by substracting the mean RGB value of the DIV2K dataset. Data augmentation includes random horizontal flips and rotations. Every weight layer in CRNet-B has 64 filters ($n_0 = 64$) with the size of $3 \times 3$ except $\boldsymbol{W}_l$ and $\boldsymbol{S}$ have $1,024$ filters ($m_0 = 1,024$). CRNet-B is updated using Adam (Kingma & Ba, 2014) with $\beta_1 = 0.9$, $\beta_2 = 0.999$, and $\epsilon = 10^{-8}$. The mini-batch size is 16. We use RGB LR patches of size $54 \times 54$ as inputs. For each epoch, there are $10^3$ iterations. The initial learning rate is $10^{-4}$ and halved every 200 epochs. We train CRNet-B for 800 epochs. Unlike CRNet-A, CRNet-B is trained using L1 loss for better convergence speed.

**Recursion** We choose $K = 25$ in both of our models. We implement our models using the PyTorch (Paszke et al., 2017) framework with NVIDIA Titan Xp. It takes approximately 4.5 days to train CRNet-A, and 15 days to train CRNet-B.

## A.5 PARAMETER STUDY

The key parameters in both of our models are the number of filters ($n_0, m_0$) and recursions $K$.

**Number of Filters** We set $n_0 = 128, m_0 = 256, K = 25$ for CRNet-A as stated in Section A.4. In Fig. 8(a), CRNet-A with different number of filters are tested (DRCN (Kim et al., 2016b) is used for reference). We find that even $n_0$ is decreased from 128 to 64, the performance is not affected greatly. On the other hand, if we decrease $m_0$ from 256 to 128, the performance would suffer an obvious drop, but still better than DRCN (Kim et al., 2016b). Based on these observations, we set the parameters of CRNet-B by making $m_0$ larger and $n_0$ smaller for the trade off between model size and performance. Specifically, we use $n_0 = 64, m_0 = 1024, K = 25$ for CRNet-B. As shown in Fig. 8(b), the performance of CRNet-B can be significantly boosted with larger $m_0$ (MDSR (Lim et al., 2017) and MSRN (Li et al., 2018) are used for reference). Even with small $m_0$, i.e., 256, CRNet-B still outperforms MSRN (Li et al., 2018) with fewer parameters (2.0M vs. 6.1M).

**Number of Recursions** We also have trained and tested CRNet-A with 15, 20, 25, 48 recursions, so the depth of the these models are 20, 25, 30, 53 respectively. The results are presented in Fig. 9(a). It's clear that CRNet-A with 20 layers still outperforms DRCN with the same depth and increasing $K$ can promote the final performance. The results of using different recursions in CRNet-B are shown in Fig. 9(b), which demonstrate that more recursions facilitate the performance improved.

## A.6 MORE QUALITATIVE RESULTS

In Fig. 10-15, we provide additional visual results on benchmark datasets to clearly show the superiority of our proposed models. The pre-upsampling models (SRCNN (Dong et al., 2016), VDSR (Kim et al., 2016a), DRCN (Kim et al., 2016b), DRRN (Tai et al., 2017a), MemNet (Tai et al., 2017b)) and the post-upsampling models (MSRN (Li et al., 2018), EDSR/MDSR (Lim et al., 2017), RDN (Zhang et al., 2018)) are used for comparisons. Red indicates the best performance.

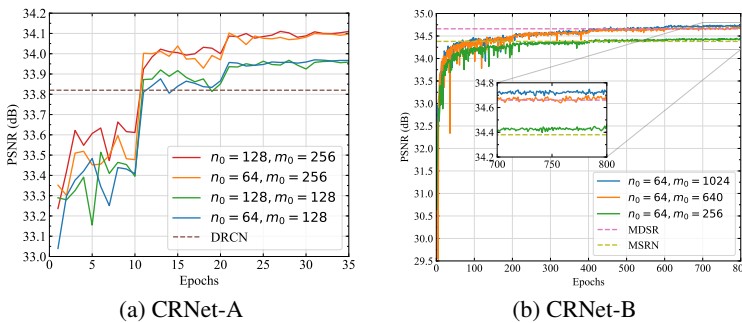

(a) CRNet-A                    (b) CRNet-B

Figure 8: PSNR of proposed models versus different number of filters on Set5 with scale factor ×3.

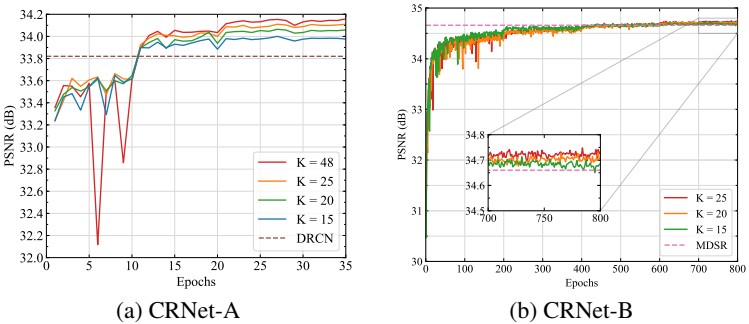

(a) CRNet-A                    (b) CRNet-B

Figure 9: PSNR of proposed models versus different number of recursions on Set5 with scale factor ×3.

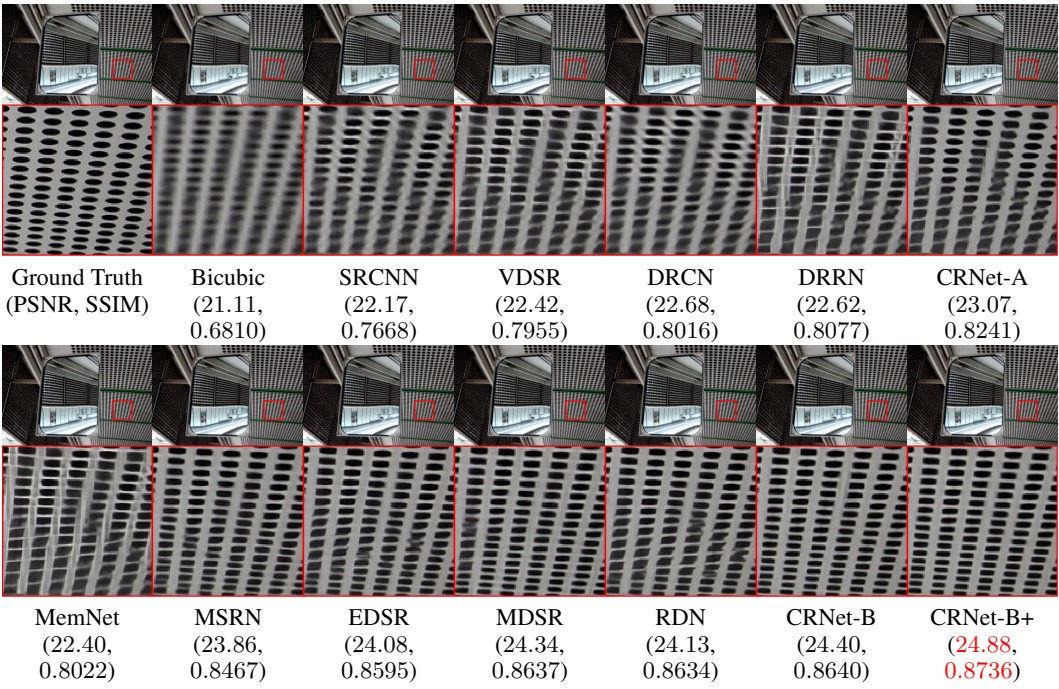

Figure 10: SR results of "img004" from **Urban100** with scale factor ×4.

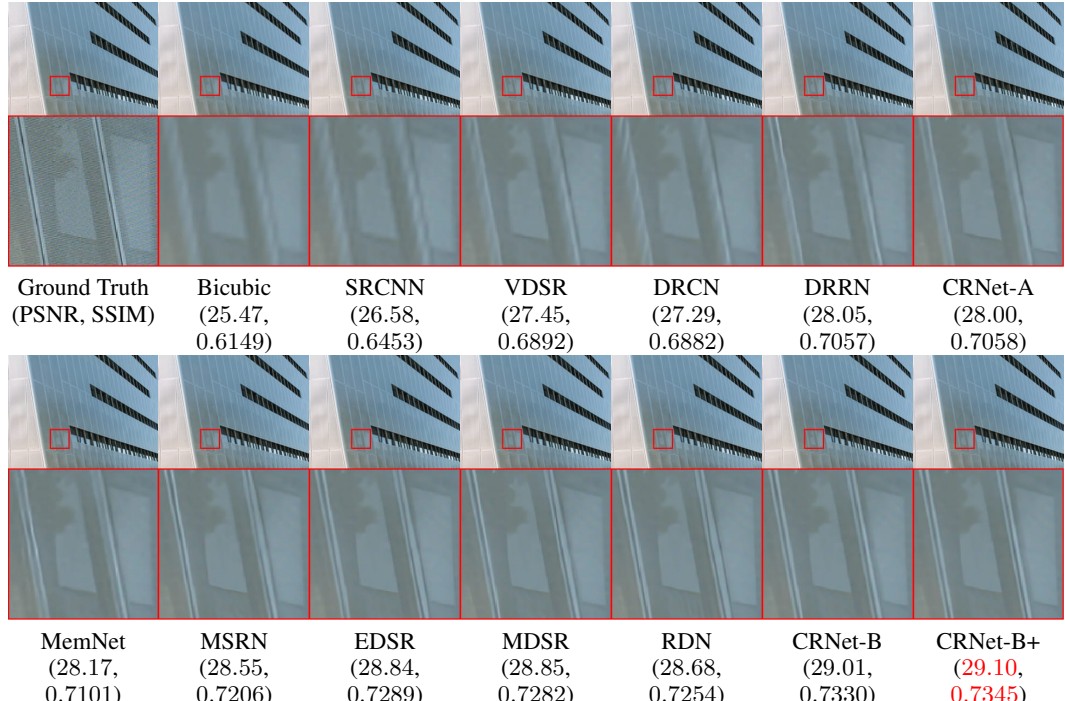

Figure 11: SR results of "img026" from **Urban100** with scale factor ×4.

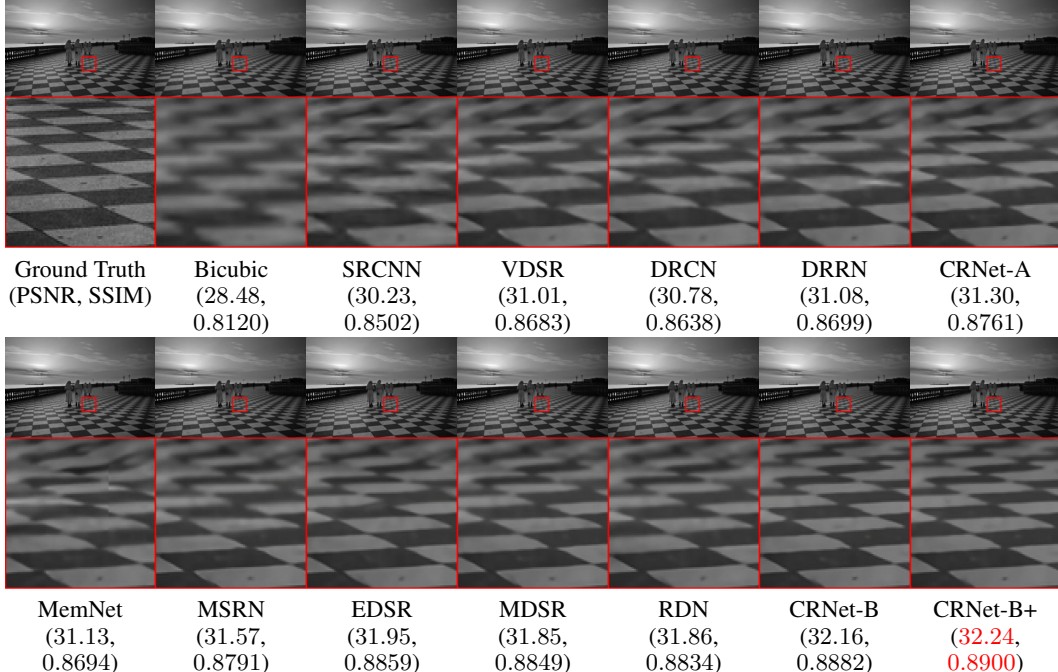

Figure 12: SR results of "img028" from **Urban100** with scale factor ×4.

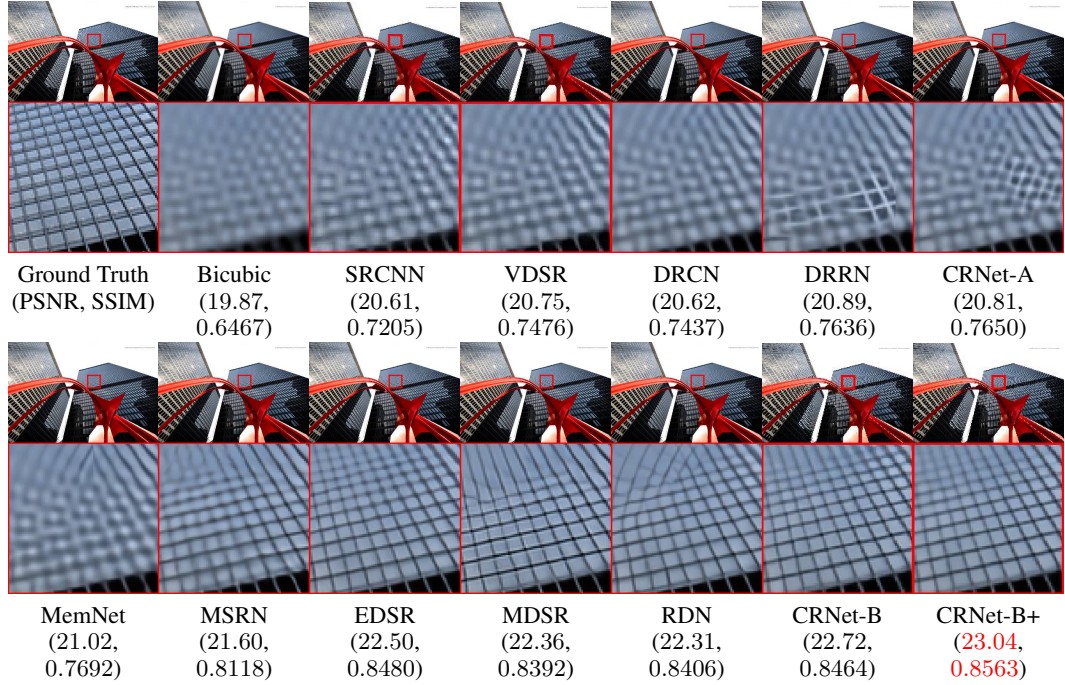

Figure 13: SR results of "img062" from **Urban100** with scale factor ×4.

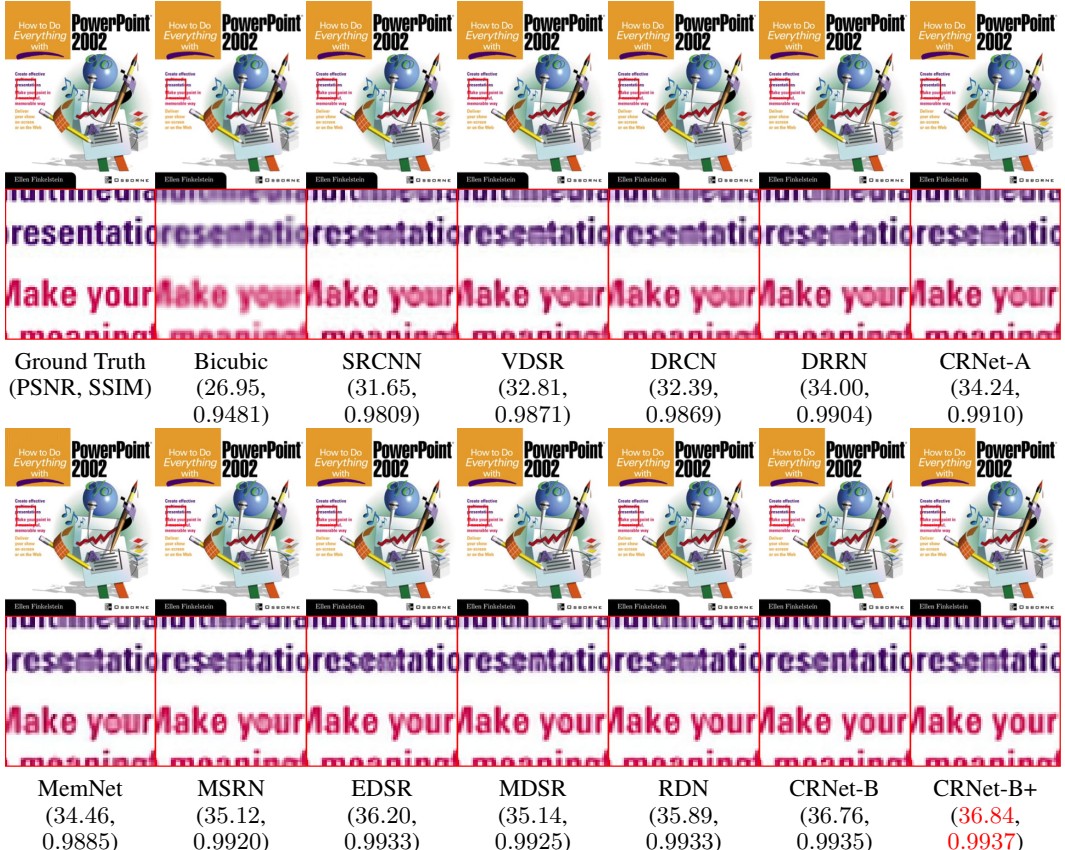

Figure 14: SR results of "ppt3" from **Set14** with scale factor ×2.

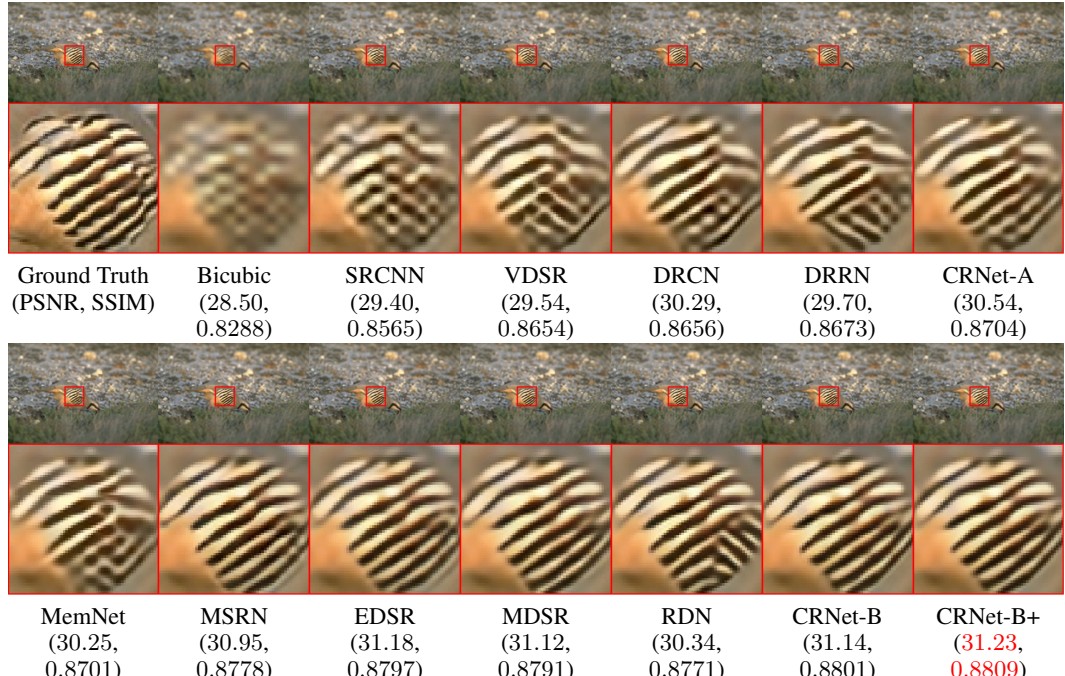

Figure 15: SR results of "8023" from **B100** with scale factor ×4.

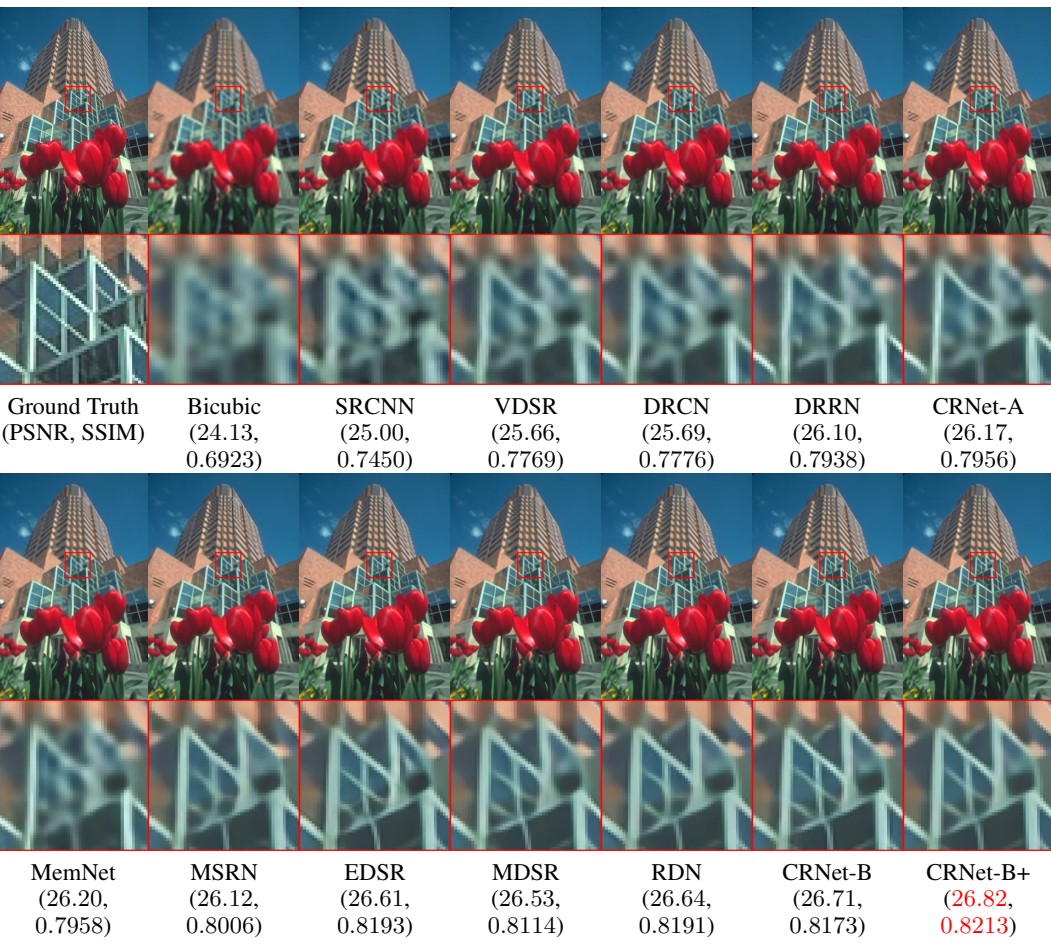

Figure 16: SR results of "86000" from **B100** with scale factor ×4.

