# OpenReview forum: "CRNet: Image Super-Resolution Using A Convolutional Sparse Coding  Inspired Network"
_ICLR.cc/2020/Conference — Reject_

### Official Review · AnonReviewer2 · 2019-10-23
**Official Blind Review #2**

**Rating:** 3

**Review:**

This work exploits the natural connection between CSC and Convolutional Neural Networks (CNN) to address CSC based image SR. Specifically, Convolutional Iterative Soft Thresholding Algorithm (CISTA) is introduced to solve CSC problem and state-of-the-art performance is achieved on popular benchmarks.

[Strengths]
- This paper is well-written and easy to follow.

[Weaknesses]
- My main concern on this work is its novelty seems to be limited. In Introduction, the authors raised 4 kinds of issues, including framework/optimization/memory/multi-scale. However, optimization and memory issues are mitigated by the CNN architectures; multi-scale issue is addressed with the help of work from Kim et al.. It seems the proposed questions are mostly addressed by the previous methods.

- It seems CISTA may be the main novelty of this work. However, no comparasions with previous ISTA methods (e.g., [R1-3]) are conducted and discussed.

- From Tabs. 2&3, the improvement is very limited. Besides, time complexity needs to be compared with the competitors, including flops or inference times.

[R1] A Fast Proximal Method for Convolutional Sparse Coding, IJCNN 2013.
[R2] Convolutional Neural Networks Analyzed via Convolutional Sparse Coding, JMLR 2017.
[R3] On Multi-Layer Basis Pursuit, Efficient Algorithms and Convolutional Neural Networks, TPAMI 2018.

**Experience Assessment:**

I have published in this field for several years.

**Review Assessment: Checking Correctness Of Derivations And Theory:**

I assessed the sensibility of the derivations and theory.

**Review Assessment: Checking Correctness Of Experiments:**

I carefully checked the experiments.

**Review Assessment: Thoroughness In Paper Reading:**

I read the paper thoroughly.

---

### Official Review · AnonReviewer1 · 2019-10-23
**Official Blind Review #1**

**Rating:** 1

**Review:**

This work investigates the problem of super-resolution using
convolutional sparse coding (CSC). The work is motivated
by a reduction in optimization time stating that the
state-of-the-art CSC solvers are slow and require to
put the entire image in memory. This is not true,
there exist online approaches such as https://arxiv.org/abs/1706.09563
or distributed methods https://arxiv.org/abs/1901.09235.

The use of an unfolding algorithm similar to LISTA is
not clearly motivated and explained, especially concerning
the training procedure when working in the SR context.

Finally the latex formatting suffers from many issues and typos.
Be careful with difference between \citet and \citep for citations.

Fonts in figure 4 are too small


**Experience Assessment:**

I have published one or two papers in this area.

**Review Assessment: Checking Correctness Of Derivations And Theory:**

N/A

**Review Assessment: Checking Correctness Of Experiments:**

I assessed the sensibility of the experiments.

**Review Assessment: Thoroughness In Paper Reading:**

I read the paper at least twice and used my best judgement in assessing the paper.

---

### Official Review · AnonReviewer3 · 2019-10-24
**Official Blind Review #3**

**Rating:** 1

**Review:**

This paper presents a single image super-resolution method. The discussion refers to sparse coding as motivation and how such coding can be achieved within a neural network using activation functions, and then it slides into iterative soft thresholding and ends up with the observation that ReLU and the soft nonnegative thresholding operator are equal (paper should have pointed that they are equal with slight difference around the bias term). This argument is not new, this conclusion has already been known and analyzed extensively in (Papyan et al., 2017a).

The proposed network is not much different from what several other SR methods used before (see NTIRE 2019). It is basically composed of a collection of single layer residual units that share the same parameters. As expected, there is also a skip connection from the input to the last layer. The only slight variation is that the activation function ReLU for the residual layers are arranged before the convolutional layers.

Training details are missing.

The paper does not provide any comparisons with the top-ranking methods in the NTIRE 2019 single image super-resolution challenge leaderboard.

**Experience Assessment:**

I have published in this field for several years.

**Review Assessment: Checking Correctness Of Derivations And Theory:**

I carefully checked the derivations and theory.

**Review Assessment: Checking Correctness Of Experiments:**

I carefully checked the experiments.

**Review Assessment: Thoroughness In Paper Reading:**

I read the paper thoroughly.

---

### Decision · Program_Chairs · 2019-12-19

**Decision:**

Reject

**Comment:**

All three reviewers agreed that the paper should not be accepted. No rebuttal was offered, thus the paper is rejected.